# Granular Insights: Neutrophil Predominance and Elastase Release in Severe Asthma Exacerbations in a Pediatric Cohort

**DOI:** 10.3390/cells13060533

**Published:** 2024-03-18

**Authors:** Kirstin Henley, Erin Tresselt, Jessica S. Hook, Parth A. Patel, Michelle A. Gill, Jessica G. Moreland

**Affiliations:** 1Department of Pediatrics, University of Texas Southwestern Medical Center, Dallas, TX 75390, USA; kirstin.henley@bcm.edu (K.H.); erin.tresselt@utsouthwestern.edu (E.T.); jessica.hook@utsouthwestern.edu (J.S.H.);; 2Department of Microbiology, University of Texas Southwestern Medical Center, Dallas, TX 75390, USA

**Keywords:** asthma, neutrophil, eosinophil, elastase

## Abstract

The chronic inflammatory component of asthma is propagated by granulocytes, including neutrophils and eosinophils, in the peripheral circulation and airway. Previous studies have suggested that these cells have an altered expression of adhesion-related molecules and a propensity for the release of granule contents that may contribute to tissue damage and enhance inflammatory complications in patients with status asthmaticus. The goal of this prospective cohort study at a tertiary care pediatric hospital with a large population of asthma patients was to assess the role of granulocyte-based inflammation in the development of asthma exacerbation. Subjects were enrolled from two patient populations: those with mild-to-moderate asthma exacerbations seen in the emergency department and those with severe asthma admitted to the intensive care unit (PICU). Clinical data were collected, and blood was drawn. Granulocytes were immediately purified, and the phenotype was assessed, including the expression of cell surface markers, elastase release, and cytokine production. Severe asthmatics admitted to the PICU displayed a significantly higher total neutrophil count when compared with healthy donors. Moreover, little to no eosinophils were found in granulocyte preparations from severe asthmatics. Circulating neutrophils from severe asthmatics admitted to the PICU displayed significantly increased elastase release ex vivo when compared with the PMN from healthy donors. These data suggest that the neutrophil-based activation and release of inflammatory products displayed by severe asthmatics may contribute to the propagation of asthma exacerbations.

## 1. Introduction

Asthma is one of the most common chronic childhood diseases in industrialized countries [1] and is recognized as a heterogenous chronic inflammatory disorder of the airways, characterized by recurrent episodes of wheezing associated with widespread airflow obstruction [2,3]. The pathogenesis triad includes bronchoconstriction, airway hyperresponsiveness, and airway edema caused by inflammation [4]. The diagnosis of asthma encompasses a broad spectrum of disease severity. Severe asthma has most commonly been differentiated by the inability of inhaled corticosteroids and additional medication to control symptoms [3]. Patients with severe asthma requiring pediatric intensive care unit (ICU) admission form a significant component of the morbidity and mortality from this disease [5], accounting for approximately 500,000 admissions to the pediatric ICU on an annual basis and 322 deaths in patients less than 25 years old in 2014 alone [6,7]. Although the majority of asthmatic patients have effective treatment options, a substantial subset of patients continue to have severe asthma that is difficult to treat [3]. These patients are an understudied population, and the mechanisms differentiating severe asthma pathogenesis are largely unknown [8].

Previous studies have revealed that there are distinct endotypes of asthma focused on the CD4+ T-cell responses, classified as type 2 (T2)-high endotype vs. non-T2 or T2-low asthma [9]. However, the expanding literature has focused on the role of specific granulocyte subpopulations, including neutrophils and eosinophils, and the phenotyping of cell surface receptors and cytokine production in asthmatics [10,11]. Despite ongoing work, there are limited data on the subgroup of severe asthmatics requiring ICU management and minimal investigation of the most critically ill pediatric asthma patients. The inflammatory component of the asthma pathophysiology triad plays an integral part in this disease and has been proposed to be driven by the activation of neutrophils and eosinophils [11].

Neutrophils and eosinophils are recruited to the airway and have been reported as critical contributors to severe asthma exacerbations [4]. The chemokine expression of both of these cell types play an integral role in the pathology of asthma and can be used as therapeutic biologic targets [12]. IL-5 plays an important role in the activation and proliferation of eosinophils. Eosinophils display enhanced abundance of IL-5R mRNA in the bronchial biopsies of atopic and non-atopic asthmatic subjects. Given these findings, novel drugs that inhibit IL-5 action have been approved for use in severe eosinophilic asthmatics [13]. Neutrophils are now recognized to synthesize and secrete cytokines implicated in the severity of asthma exacerbations, namely CXCL8 (IL-8). This cytokine was analyzed in the tracheal aspirates of severe asthmatics and was found to correlate significantly and positively with the length of the asthma exacerbation [14]. Moreover, there is a growing body of studies demonstrating a correlation with neutrophilia and severity of disease [15]. We hypothesized that there would be unique and distinct phenotypes of circulating neutrophils and eosinophils when comparing samples from children with severe asthma requiring ICU admission vs. mild or moderate asthmatic children who are seen in the emergency room (ER) setting vs. healthy donors. Activation of both neutrophils and eosinophils may contribute to more severe asthma exacerbations, secondary to their ability to cause significant host tissue damage [16,17].

In the current study, we measured several well-described markers of neutrophil activation, including the assessment of cytokine and elastase release from granulocytes isolated from the peripheral blood of severe asthmatics requiring PICU admission and ER patients. We also compared the aforementioned groups with healthy donors. Activation markers of eosinophils in the peripheral blood, including CD125, CD69, and CD193, from moderate asthma patients were compared with healthy donors, but there were insufficient eosinophils in the blood of severe asthmatics for analysis. Intriguingly, we found that neutrophil elastase secretion was upregulated in severe asthmatics as compared with healthy donors and might be a critical component in the propagation of inflammation in this subgroup of patients.

## 2. Materials and Methods

### 2.1. Study Subjects

This prospective cohort study took place in an urban, academic, and tertiary care PICU and ER. Patients were enrolled between August 2018 and August 2022 (excluding a break in enrollment due to COVID-19). Patients aged 2–17 years with status asthmaticus that were admitted to the PICU or seen in the ER were approached for inclusion in the study. Severe asthma exacerbation was defined as requiring PICU admission due to the necessity of continuous nebulization treatments or requirement for bilevel positive airway pressure (BiPap). Subjects with known or documented viral infections were also included, as viral infection is a common cause for asthma exacerbation. Though not excluded from the study, no patients diagnosed with COVID-19 were enrolled. Exclusion criteria included patients outside of the abovementioned age groups and those with accompanying underlying inflammatory conditions (e.g., inflammatory bowel disease, oncological diagnosis, etc.). If both parties agreed to participation, parents of children in the study signed written informed consent at the time the samples were obtained, and subjects greater than 10 years and less than 18 years were asked for assent. The UT Southwestern Medical Center Institutional Review Board and Children’s Medical Center in Dallas approved this prospective minimal risk cohort study, and informed consent was obtained from each donor.

### 2.2. Data Collection

At the time of blood collection, data were obtained from the patient’s electronic medical record, including demographic data (age, race, ethnicity, sex), asthma history including type of asthma, previous hospital or PICU admissions for asthma, viral positivity, and comorbidities such as allergies and eczema.

### 2.3. Blood Collection and Human Neutrophil and Eosinophil Purification

A 10 mL sample of peripheral blood was obtained from an existing line or via venipuncture, placed directly into a lithium heparin tube using universal precautions, and taken to the lab for immediate processing. Samples were collected within 24 h of PICU admission or within 2 h of presentation to the ER after informed consent and assent, when appropriate, were obtained in accordance with an IRB approved protocol (STU 032018-076). Granulocytes were isolated using dextran sedimentation followed by a density gradient using Ficoll-Hypaque as previously described [18]. The isolation was then completed with erythrocyte lysis using hypotonic solutions. A manual differential was performed to assess granulocyte purity (99.39 ± 0.89%, 99.67 ± 0.50%, and 97.27 ± 2.01% for PICU, ER, and healthy donors, respectively) and neutrophil and eosinophil percentage (Figure 1).

### 2.4. Cell Surface Expression of Neutrophil and Eosinophil Markers of Activation

Flow cytometry was utilized to identify cell surface protein expression associated with cellular activation. CD11b (BD 555386), L-selectin (BD 555542), and ICAM-1 (BD 555510) were measured in neutrophils, and CD125 (IL5Ra) (BD 555902), CD69 (BD 560738), and CD193 (BD 564189) were measured in eosinophils. The cells were fixed with 4% formaldehyde for 30 min on ice followed by staining on ice. Stained cells were resuspended in PBS and analyzed by flow cytometry using a FACSCalibur from BD Biosciences (San Jose, CA, USA) in the Flow Cytometry Facility at the University of Texas Southwestern Medical Center with consistent voltage settings for all samples. Neutrophils and eosinophils were identified by forward- and side-scatter techniques (neutrophils) and Siglec-8 (eosinophils) (Miltenyi Biotech, Gaithersburg, MD, USA, 130-098-729), and the geometric mean intensity was determined using Flowjo analysis software, version 10.0.08, from TreeStar (Ashland, OR, USA).

### 2.5. Ex Vivo Cytokine Production and Elastase Release

Isolated granulocytes underwent a 20 h incubation in RPMI and 10% pooled human serum in a tissue culture incubator with constant rotation, after which cell-free supernatant was harvested and stored at −80 °C until ELISA was performed. Neutrophil azurophilic granule mobilization as estimated by elastase release was measured by ELISA using the PMN (neutrophil) Elastase Human ELISA Kit (Invitrogen BMS269, Carlsbad, CA, USA) as per the manufacturer’s instructions. IL-6 and IL-8 production and release were measured by ELISA using the Human IL-6 ELISA Kit (Invitrogen EH2IL6, Carlsbad, CA, USA) and Human IL-8 ELISA Kit (Invitrogen 88-8086-88, Carlsbad, CA, USA), respectively, as per the manufacturer’s instructions. Patient and healthy donor control samples were included on the same plate to reduce plate-to-plate variability. Absorbance at 450 nm was measured on a Clariostar Omega from BMG Labtech (Cary, NC, USA).

### 2.6. Statistical Analysis

Data collection included (1) demographic data, including age, sex, race, and ethnicity; (2) asthma history, including severity of asthma and comorbidities such as allergies and eczema and current medications; and (3) admission data, including current PICU admission/ER visit and viral positivity. Data were analyzed using GraphPad Prism software version 10.0.1 (La Jolla, CA, USA). We assessed the density of six specific cell surface receptors (three on neutrophils and three on eosinophils) known to be upregulated during asthma exacerbations. Student’s *t*-tests and one-way analysis of variance (ANOVA) tests with multiple comparisons were used where appropriate. Results were considered significant with *p*-value ˂ 0.05. * *p* ˂ 0.05, ** *p* ˂ 0.01, *** *p* ˂ 0.001, and **** *p* ˂ 0.0001. Baseline and demographic characteristics were summarized using standard descriptive measures: means with standard deviation (SD) for normally distributed continuous variables, medians with interquartile ranges (IQR) for continuous variables that are not normally distributed, and percentages for categorical variables.

## 3. Results

### 3.1. Enrollment

A total of 42 study subjects were enrolled at the UT Southwestern Medical Center and Children’s Medical Center in Dallas, TX, between 24 August 2018 and 29 August 2022. Twenty-three study subjects were admitted to the PICU for severe asthma exacerbations requiring continuous nebulization treatments and additional ICU therapies, including non-invasive or invasive mechanical ventilation and terbutaline infusions. Nine study subjects were evaluated in the ER for mild-to-moderate asthma exacerbations. Ten healthy donors were enrolled as controls. Table 1 provides the overall demographics of patients and healthy donors, as well as clinical variables for patients. Although all PICU patients received intravenous steroids, 21 of 23 PICU patients received corticosteroids prior to blood collection. The average time to blood draw for this population was 13.1 h post first steroids. In the mild-to-moderate asthma subgroup encountered in the emergency department, all of them received steroids with a mean time of 1.3 h prior to blood sampling. In this group, four patients received oral prednisone and the remainder received oral dexamethasone in the emergency department. For the severe asthma group, all patients were started on IV methylprednisone in the ED or in the PICU. There were no significant differences in home asthma regimens between the cohorts (Appendix A). All subjects reported having an albuterol inhaler at home, and 11/23 PICU patients had some form of inhaled steroid therapy prescribed prior to this visit. Most of the patients on inhaled corticosteroids had a twice daily dosing prescribed, although the actual frequency of use was not known. Another three patients had nasal steroid use prescribed, Flonase, for allergic rhinitis. In the ED group, three of nine patients had Flovent prescribed in addition to albuterol. None of the subjects had biologic therapy prescribed for eosinophilic asthma. Immunoglobulin (IgE) levels had been measured in a minority of both groups and were not statistically different with a wide range of levels (Table 1). The average BMI of the PICU cohort was in the overweight range. Although the sample size was too small to make BMI comparisons between the mild-to-moderate group and the PICU cohort, the finding of high BMI in severe asthma cohort was not surprising given the published association between greater asthma symptoms and overweight BMI [19].

### 3.2. Neutrophilia in Children with Severe Asthma Exacerbations

Using granulocyte purification, we found that patients admitted to the PICU with severe asthma had 3.9-fold more neutrophils in circulation than healthy donors and a trend towards increased neutrophils compared with the patients in the ER (Figure 1A). Intriguingly, despite the paradigm of eosinophil involvement in severe asthma, pediatric ICU patients with severe exacerbations had little to no eosinophils in the granulocyte preparation (Figure 1B,C). Although not all subjects had blood drawn and sent to the clinical lab for white cell measurements, 10 of our 23 ICU patients did have a complete blood count with differentials. Consistent with what was noted in the granulocyte preparation from the laboratory, the patients had very low or unmeasurable eosinophil counts, with a mean absolute eosinophil count of 0.04 (thousand/mm^3^). Of the PICU patients with complete blood count with differentials, 60% had zero eosinophils. The inflammatory phenotype of granulocytes isolated from patients admitted to the PICU versus ER was assessed. Due to the limited number of eosinophils isolated, eosinophil phenotype was not assessed for patients admitted to the PICU.

### 3.3. Neutrophil and Eosinophil Inflammatory Cell Surface Markers

Intercellular adhesion molecule 1 (ICAM-1) is a cell surface glycoprotein best known for its function in endothelial and epithelial cells. ICAM-1 binds to integrins, including the β-2 integrin CD11b, on the surface of cells of the immune system and facilitates extravasation of leukocytes from the bloodstream into tissue [20]. ICAM-1 is not constitutively expressed on neutrophils, but it has recently been recognized that expression can be induced by a variety of inflammatory stimuli [21]. Using the flow cytometry of freshly isolated neutrophils, we found a trend towards increased ICAM-1 expression on neutrophils from patients with severe acute asthma exacerbations as compared with healthy donors, though the difference was not statistically significant. Interestingly, neutrophils isolated from the peripheral blood of asthma patients seen in the ER with mild-to-moderate exacerbations expressed the greatest amount of ICAM-1 (Figure 2A). CD11b is constitutively expressed on neutrophils and is rapidly upregulated under inflammatory conditions [22]. Intriguingly, we observed no difference in surface CD11b expression between the three cohorts (Figure 2B). L-selectin is a cell adhesion molecule that is constitutively expressed on the surface of most leukocytes. It plays a central role in the initial binding and rolling of cells on the endothelium [23]. Shedding of L-selectin, a classic hallmark of neutrophil activation, can be initiated by interaction with the endothelium or exposure to soluble inflammatory stimuli [23]. We observed enhanced L-selectin shedding in neutrophils isolated from the peripheral blood of patients admitted to the PICU in comparison with healthy donors (Figure 2C).

Given the well-described role of eosinophils in asthma, we also analyzed the surface expression of three markers of inflammation on the surface of eosinophils. As mentioned above, patients treated in the PICU with severe exacerbations did not have adequate numbers of eosinophils for analysis (Figure 1C). CD125 is a receptor for IL-5, a key mediator of allergic diseases, including allergic rhinitis and asthma. Mepolizumab and reslizumab, monoclonal antibodies that target IL-5, are used to treat asthma. CD69 is a marker of eosinophil activation and has been correlated with clinical findings. CD193 is a receptor for eotaxin, an eosinophil chemoattractant [24]. There were no significant differences in the surface expression of CD69 or CD193 on eosinophils from the peripheral circulation of patients with mild-to-moderate asthma versus healthy donors. However, eosinophils from patients with mild-to-moderate asthma displayed greater expression of CD125 than eosinophils from healthy donor controls (Figure 3).

### 3.4. Elastase Secretion by Neutrophils during Asthma Exacerbation

Given the absence of eosinophils and the suggestion that the neutrophils from severe asthma patients display an activated phenotype, we focused specifically on the ICU population and measured secreted inflammatory products that might enhance the lung inflammation that accompanies severe asthma. Using an ex vivo culture model, we measured secreted elastase from isolated neutrophils. Granulocytes isolated from the peripheral blood of patients with severe asthma exacerbations released more elastase than granulocytes isolated from healthy donors (Figure 4).

### 3.5. Cytokine Secretion by Neutrophils

We next studied secreted inflammatory cytokines produced by activated neutrophils. It is now recognized that IL-6 and IL-8 can be synthesized by neutrophils. IL-6 is a neutrophil chemoattractant and has numerous well-characterized pro- and anti-inflammatory functions [25,26,27]. Although there was wide variability between donors, there was a trend towards enhanced cytokine production from cells from PICU asthma patients as compared with those isolated from healthy donors (Figure 5A,B). The IL-8 levels in particular appeared to have a distinct population of high producers that might reflect a subgroup of patients with more extensive neutrophilic infiltration into the lung. A larger sample size will be needed to see if these distinct subgroups persist.

## 4. Discussion

Asthma is defined by the symptoms of bronchoconstriction, airway hyperresponsiveness, and airway edema, caused by inflammation. The inflammatory component of asthma is driven by specific cell mediators, namely neutrophils and eosinophils [28,29]. When these cells are activated, they have the propensity to cause significant host tissue damage secondary to the release of cytokines and proteolytic enzymes [14]. Severe asthma is a significant cause of morbidity and mortality in the pediatric population. Asthma exacerbations account for approximately 500,000 admissions to pediatric ICUs and hundreds of pediatric deaths per year in the U.S. [30]. Critically ill pediatric patients are an understudied population, and the cellular mechanisms underlying severe asthma are not well described. Given that the care of asthmatics continues to be highly algorithmic, we sought to explore the distinguishing phenotypes of neutrophils and eosinophils to assist in the discovery of more individualized care.

The current study demonstrates two novel findings. First, pediatric ICU patients admitted with severe asthma exacerbation have a paucity of eosinophils (unmeasurable in majority of the patients) and enhanced neutrophil numbers as compared with both mild-to-moderate asthma patients and healthy donors. Moreover, these neutrophils are activated and release greater amounts of elastase when cultured ex vivo. Elastase is a proteolytic enzyme packaged in the primary or azurophilic granules of neutrophils during hematopoiesis and maturation of neutrophils in the bone marrow. Primary granules contain the most toxic granule contents that are released into the phagosome during phagocytosis for pathogen killing. The release of elastase extracellularly has been implicated in acute lung injury and has been therapeutically targeted [31,32].

Our assay for the quantification of extracellular elastase would not distinguish between secreted elastase, i.e., regulated exocytosis, and the extrusion in neutrophil-extracellular traps or NETs. Recently, there has been more interest in the role of NETs in the pathogenesis of asthma [33]. The role of NETs in pathogen killing is clearly necessary in some asthma exacerbations, but NETs can also be induced in viral settings, including in response to rhinovirus and influenza [34,35], common triggers for asthma exacerbation in children. NETs have been identified in bronchial washings [36] and, importantly, plasma measures of NETs have been associated with disease severity in asthma [37]. Taken together, a more directed analysis of NET generation potential in neutrophilic asthma is warranted to develop precision therapies.

Evidence for neutrophil-predominant asthma as a distinct phenotype has gained traction. While the presence of neutrophils was previously described as a consequence of therapy, particularly corticosteroid use, there is evidence of neutrophil predominance in a subset of asthma patients, even in the absence of steroid therapy. The association between neutrophilic inflammation and disease severity in asthma has been demonstrated both in adult populations [15] and in pediatric studies [38]. These findings have led to attempts to identify biomarkers for neutrophil-predominant asthma using a bioinformatics approach [39].

Our study sought to measure cell surface proteins that are well-described markers of neutrophil activation. L-selectin is a cell adhesion molecule that is constitutively expressed on the surface of most leukocytes. It plays a central role in the initial binding and rolling of cells on the endothelium [23]. Shedding of L-selectin, a classic hallmark of neutrophil activation, can be initiated by interaction with the endothelium or exposure to soluble inflammatory stimuli [40]. We did find increased L-selectin shedding in the PICU cohort as compared with healthy donors. We did not find a statistically significant difference in the surface expression of ICAM-1, although the trend suggested enhanced ICAM-1 in both asthma subgroups as compared with the control. Further study of ICAM-1 regulation in neutrophils during asthma is warranted given the critical role that ICAM-1 plays as a receptor for adhesion for a subset of rhinoviruses and thus facilitates entry and may propagate disease. In our PICU population, 14 of 23 patients tested positive for rhinovirus. Recent work has been directed at anti-ICAM-1 therapies to reduce viral entry and downregulate inflammation [41].

Given the lack of eosinophils in the severe asthma patients, no measure of cell activity in this subgroup could be obtained. One potential explanation for this finding is the relationship with corticosteroid dosing. Corticosteroids have been demonstrated to reduce blood eosinophil counts in chronic use [42], although an acute impact of eosinophil count in asthma has not been specifically reported. One of the markers of eosinophil activation, CD125 expression, was enhanced in cells from mild-to-moderate asthma seen in the ER compared with healthy controls, but other markers were equivocal. Previous studies have shown an upregulation of these markers during asthma exacerbations. It is possible that eosinophils with a highly activated phenotype are either marginated in the lung or already extravasated by the time we see the patient in the ER with an exacerbation. Moreover, the ER donor number was low, which could impact the power of these results.

Our studies on neutrophil cytokines did not provide conclusive data. Healthy donor IL-6 and IL-8 are generally undetectable (IL-6) or very low (IL-8) using our in vitro assay. Given the activated neutrophil phenotype, the very low IL-8 levels were striking in severe asthma patients. The acute inflammatory response is a dynamic process with a mix of both pro- and anti-inflammatory responses. Our timing for sampling after admission to the ICU was within the first 24 h; thus, this range may not have sampled the peak neutrophil activation/production of cytokines. The low levels of IL-8 may also have been a result of release of relatively “immature” neutrophils from the bone marrow. The exceedingly low levels of IL-6 are not surprising in the severe asthma group given the neutrophil predominance. IL-6 has not been demonstrated to be a useful biomarker for disease severity in a very large cohort of urban asthma patients [43]. Importantly, our reductionist approach with an ex vivo analysis of only granulocyte cytokine generation excludes the inflammatory status of the monocyte population. Additionally, we did not address the dendritic cell contributions, a population with a critical role in childhood asthma [44]. Moreover, the crosstalk with adaptive immunity, including a recently described role for Th17- and Th-2-mediated immunity in the pathogenesis of neutrophilic asthma in children, was not probed in our pediatric asthma cohorts [45].

There are several limitations to this study. The sample size was small, with 23 PICU patients, 9 ER patients, and 10 healthy donors, which limits the power of this study. In addition, the control population was primarily adult healthy donors, with only a few age-matched controls. The age range for the healthy donors was 10–48 years, with a median age of 31.5 years, which is much higher than the PICU and ER cohorts. At this time, there is no evidence that neutrophilic or eosinophilic inflammation differs based on age [46], which leads us to believe that the age difference among the cohorts is inconsequential. Importantly, our healthy donor cohort was also not matched for race, with only 10% of patients being Black, non-Hispanic subjects, whereas the asthma cohorts had 65% and 78% Black, non-Hispanic subjects in the mild-to-moderate and severe asthma groups, respectively. This could impact our capacity to draw any broad conclusions given the enhanced prevalence of asthma in U.S. Black, non-Hispanic children and the enhanced mortality of asthma in this subgroup [47]. In addition, our ER vs. PICU samples were not matched in terms of time since corticosteroids and dosing. Due to the small sample size, there are insufficient patient clinical data for some comparisons.

## 5. Conclusions

The current study furthers our understanding of the granulocyte phenotype during asthma exacerbation in a pediatric cohort and highlights the need for further investigation of neutrophil-based inflammation in pediatric patients requiring admission to the PICU. Importantly, we measured greater amounts of elastase, a proteolytic enzyme that has been demonstrated to damage lung tissue, in the culture supernatant of granulocytes from pediatric patients with severe asthma exacerbation, demonstrating a systemic activation of granulocytes among the most critically ill asthmatics. The findings described here suggest a potential role for targeted anti-inflammatory therapy for neutrophilic asthma. In addition, taken in the context of the current asthma literature, these data underscore the need for a broad examination of the immunophenotype of all immune cells during acute severe asthma exacerbation to generate targeted acute therapies rather than algorithmic care.

## Figures and Tables

**Figure 1 cells-13-00533-f001:**
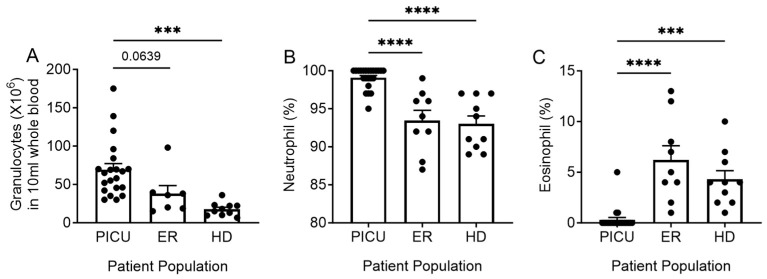
(**A**) The absolute granulocyte count is 3.9-fold (PICU) and 2.16-fold (ER) higher in the peripheral blood of patients versus healthy donors. The percentage of neutrophils (**B**) and eosinophils (**C**) in the peripheral blood of patients versus healthy donors. *** *p* < 0.001, **** *p* < 0.0001. Each dot represents a unique donor. N = 21–23 PICU; N = 7–9 ER; N = 9–10 HD.

**Figure 2 cells-13-00533-f002:**
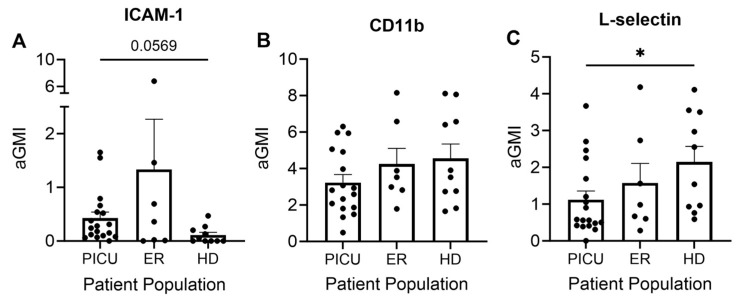
(**A**) There is a trend toward increased ICAM-1 expression on circulating neutrophils of PICU patients versus healthy donors. (**B**) There is no difference in CD11b expression among the three groups. (**C**) Enhanced L-selectin shedding in the circulating neutrophils of PICU patients versus healthy donors. * *p* < 0.05. Each dot represents a unique donor. N = 17–18 PICU; N = 7 ER; N = 10 HD.

**Figure 3 cells-13-00533-f003:**
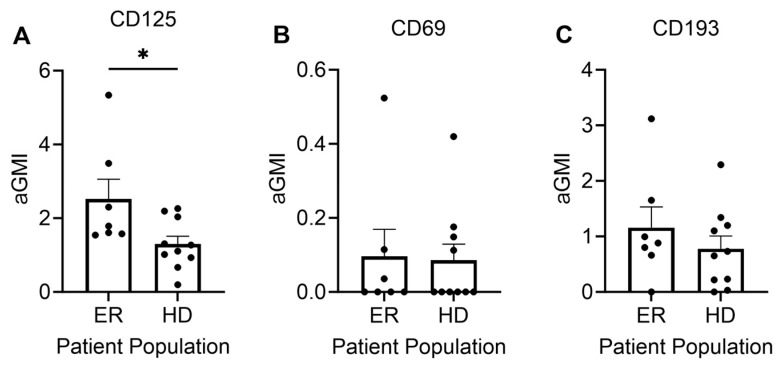
(**A**) CD125 but not CD69 (**B**) or CD193 (**C**) was increased in circulating eosinophils from ER patients versus healthy donors. * *p* < 0.05. Each dot represents a unique donor. N = 7 ER; N = 10 HD.

**Figure 4 cells-13-00533-f004:**
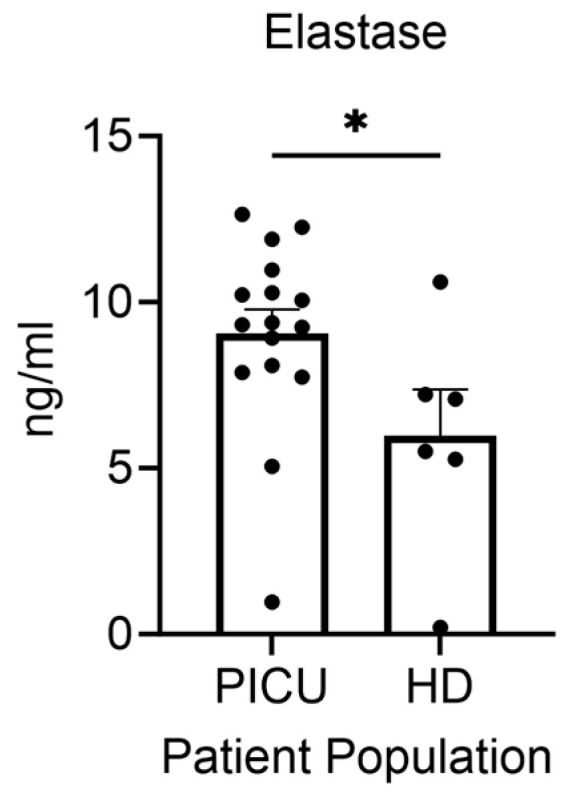
Granulocytes from pediatric patients with severe asthma exacerbations release greater amounts of elastase than granulocytes from healthy donors. * *p* < 0.05. Each dot represents a unique donor. N = 16 PICU; N = 6 HD.

**Figure 5 cells-13-00533-f005:**
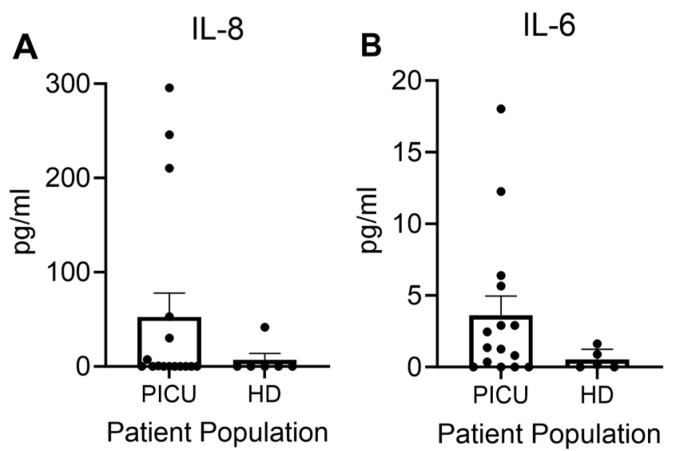
Trend toward increased (**A**) IL-8 and (**B**) IL-6 production in granulocytes from patients with severe asthma exacerbations compared with healthy donors. Each dot represents a unique donor. N = 16 PICU; N = 6 HD.

**Table 1 cells-13-00533-t001:** Demographics and clinical variables.

Variables	PICU Median (IQR) N = 23	ER Median (IQR) N = 9	Healthy Donors Median (IQR) N = 10
Age (yrs)	9 (6–13)	11 (6–13)	31.5 (15–39.75)
Sex (% male)	56.5%	78%	40%
Ethnic minority (%)	82.5%	100%	40%
Black, non-Hispanic	65%	78%	10%
White, non-Hispanic	17.5%	0%	60%
Hispanic	17.5%	22%	10%
Asian	-	-	20%
Severity of Asthma			
Mild persistent	13%	33%	
Moderate persistent	52%	22%	
Severe persistent	35%	44%	
Rhino/Entero +	67%	-	
Comorbidities:			
Allergic Rhinitis	87%	78%	
Eczema	26%	44%	
BMI (percentile)	85th (60–95) *n* = 19	42nd (21–67) *n* = 4	
IgE (IU/mL)	948 (264–2080) *n* = 11	954 (434–2136) *n* = 4	

## Data Availability

The data presented in this study are available on request from the corresponding author.

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
