# Peer review of "Granular Insights: Neutrophil Predominance and Elastase Release in Severe Asthma Exacerbations in a Pediatric Cohort"

_cells, 2024, doi:10.3390/cells13060533_

Round 1

Reviewer 1 Report

Comments and Suggestions for Authors

In this study, Henley et al., assessed the role of granulocytic inflammation in the development of asthma exacerbations in a pediatric cohort of asthma patients under tertiary care. They assessed granulocyte phenotypes using FACs-based cell surface markers, elastase release and cytokine production. They observed severe asthmatics admitted to the PICU displayed significantly higher peripheral neutrophil counts relative to healthy volunteers, with little to no eosinophil counts observed. Neutrophils derived from PICU patients displayed heightened secretion of elastase ex vivo relative to neutrophils from healthy volunteers.

The Authors concluded that the neutrophil-based activation and release of inflammatory products displayed by severe asthmatics may contribute to the propagation of asthma exacerbations.

This is an interesting translational study suggesting that elevated neutrophil numbers and activated cells may be a prognostic indicator for pediatric asthma patients at greater risk of exacerbations. It would be an important addition to the literature in a research area that remains poorly understood ie patients with non-T2 or T2-low asthma.

The following are my comments for considerations to revise the paper and resubmit it for further review.

 Patient Demographics and Clinical Variables.

1.  Patient recruitment spanned over four years, what proportion of n=42 subjects were recruited pre COVID-19 and post COVID-19?

2. The patient groups were not similar in terms of age. The data presented indicate very few age-matched controls were used as comparator data in this study as the control population appeared to be primarily comprising of healthy adult donors. Therefore, to improve the study design, the Authors should assess more age-matched controls as a comparator in their study to make their data more meaningful and generalizable in the pediatric asthma population. 

3. Similarly, the patient groups were not similar in terms of ethnicity. Notably, there is an over-representation of the African/American asthmatic children [65% & 78%, respectively] in the PICU/ER cohort relative to the Healthy Donors that comprised nearly two-thirds of European Americans children [60%].

The importance of population differences needs to be highlighted and discussed given that in the United States, asthma disproportionately affects African American children with greater burden of morbidity and mortality. Several studies have demonstrated children from African American descent have disproportionately higher rates of asthma and worse outcomes.

Moorman et al. National Surveillance for Asthma-United States, 1980–2004. MMWR Surveillance Summary. 2007;56(8):1–54.

Table 1.

4. As this is an asthma study, could the Authors please provide additional clinical variables including lung function, total serum IgE, FeNO [ppb] and BMI ?

5. Similarly, please provide information on use of medication eg ICS dose (fluticasone equivalents/day), Biologics and antibiotics intake?

6. What about family history of asthma and Smokers in the Family?

Methodology:

1.  Whilst differential counts were performed to assess granulocyte purity (97.27-99.39%), was cell viability determined and if so how, and please state granulocyte cell viability.

2.  Related to this, for the Ex Vivo Cytokine Production and Elastase Release, it is stated that isolated granulocytes underwent a 20-hour incubation in RPMI and 10% human serum in a tissue culture incubator.

What was the rationale for the 20 hour incubation in culture period given that neutrophils are short-lived cells that undergo rapid constitutive apoptosis within several hours in vitro? Importantly, what was the cell viability after 20 hour in culture?

Results:

1. Fig 4 indicates release of Neutrophil Elastase ~8 ng/ml versus  ~5 ng/ml PICU vs HD, respectively. Whilst the levels of elastase measured is statistically significant, what would be the physiological difference during disease pathogenesis?

Also please state number of patients [n=] measured from each group in the Figure legend of such analyses.

2.  Fig 5 In measuring secreted levels of IL-8 and IL-6 from activated neutrophils from PICU patients, there seems to be a low and high subset [>100 pg/ml and >5 pg/ml, respectively]. Please discuss potential reasons for neutrophil biomarker heterogeneity in the PICU asthma patients.

 Discussion:

The Authors have highlighted the importance of activated neutrophils, elastase [CXCL8/IL-8] and NETs in the pathogenesis of asthma throughout the manuscript. Additional information would further compliment their findings. For example,

1.  Patients were recruited from urban areas. What additional information were gathered concerning their home environment, smoking, exposure to air pollution etc? Please discuss in the Discussions.

2.   Line #276; The Authors touch upon the role of NETs in the pathogenesis of asthma. Do the Authors have samples available to measure any extracellular DNA in the pediatric cohort which would further compliment their data? Of clinical relevance, raised extracellular DNA has been associated with an exacerbation-susceptible phenotype of neutrophilic asthma recently [PMID: 34082773].

3.   Line #306; ‘Given the lack of eosinophils in the severe asthma patients, no measure of cell activity in this subgroup could be obtained. One potential explanation for this finding is the relationship with corticosteroid dosing.

What about the suppressive effects of anti-allergic therapies, including Omalizumab that have been shown to reduce peripheral blood eosinophil counts?

4.  Were any patients taking this Biologics therapy [or other mAbs mentioned in Line #217], if so, please include in the patient medications in Table 1 and discuss in Discussion.

5.  Line #339: The Authors conclude with ‘The current study furthers our understanding of granulocyte phenotype during asthma exacerbation in a pediatric cohort and highlights the need for further investigation of neutrophil-based inflammation in pediatric patients requiring admission to the PICU'.

Please discuss how your findings compare to another pediatric asthma study which was published recently. Wei Q et al., 2021 Allergy Asthma Clin Immunol;17(1):4.

6. Finally, based on the Authors findings, what are the Authors views in pharmacologically targeting pediatric neutrophilic asthma? Do they feel the current data (their own and published by others eg PMID: 27574789) support the selective targeting of neutrophil-based inflammation in severe asthma?                                   

Please review published clinical findings, elaborate and discuss in Discussion.

Reviewer 2 Report

Comments and Suggestions for Authors

Asthma is an important medical and social problem, especially in children. Therefore, the study topic is of clinical and research interest.

Comments:

1.It is not clear from the description of the materials and methods whether the patients had been diagnosed with asthma previously before the current severe exacerbation and what therapy they were receiving. Were other clinical data including saturation, PaCO2, spirometry, body temperature, etc. evaluated? Were general clinical laboratory data, immunoglobulin E levels evaluated?

2 From the description of materials and methods it is not clear how the group of healthy subjects was formed. Were there smokers among them?

3. What corticosteroids did patients in the PICU group and the mild to moderate group receive?  What overall treatment were the patients receiving, including prior to hospitalization? 

4. How was the percentage of neutrophils in the fraction obtained with Hypaque-Ficoll monitored. What was the percentage of neutrophils in the fraction obtained?

5. The abstract should be made more informative by including data from materials and methods and results.

6. It is recommended that a conclusion section be added to summarize the data and describe directions for future research.

Round 2

Reviewer 1 Report

Comments and Suggestions for Authors

Following subsequent revision of the manuscript by the Authors, the individual points of feedback have been adequately addressed with their findings elaborated in a bit more detail to help the readers.

Please consider updating minor information content in Table 1 as part of minor revision.

Please provide information on use of inhaled medication ICS dose (fluticasone equivalents/day) and FeNO [ppb].

Thank you for providing the BMI information in Table 1, it is noted that the PICU patients are in the 85th percentile. By the CDC’s definition, children whose BMI fall between the 85th - 94th percentile are considered overweight.

Would the Authors discuss this in the Discussions given that data from a cross-sectional study have reported the prevalence of childhood asthma with more severe asthma symptoms amongst inner city Black and Hispanic children were significantly increased with a higher BMI of 85th percentile or greater Luder et al., J Pediatr. 1998;132(4):699-703.

Reviewer 2 Report

Comments and Suggestions for Authors

The authors added new data that improved the quality of the article. At the same time I did not find answers to some of my questions. It is recommended to add insufficient clinical data on patients to the limitation of the study.

Author Response

The authors added new data that improved the quality of the article. At the same time I did not find answers to some of my questions. It is recommended to add insufficient clinical data on patients to the limitation of the study.

Thank you for the feedback. A comment regarding this limitation has been added to the limitations paragraph in the discussion.